# Decoupled Six-Axis Force–Moment Sensor with a Novel Strain Gauge Arrangement and Error Reduction Techniques

**DOI:** 10.3390/s19133012

**Published:** 2019-07-08

**Authors:** Getnet Ayele Kebede, Anton Royanto Ahmad, Shao-Chun Lee, Chyi-Yeu Lin

**Affiliations:** 1Department of Mechanical Engineering, National Taiwan University of Science and Technology, Taipei 106, Taiwan; 2Taiwan Building Technology Center, National Taiwan University of Science and Technology, Taipei 106, Taiwan; 3Center for Cyber-Physical System, National Taiwan University of Science and Technology, Taipei 106, Taiwan

**Keywords:** six-axis force–moment sensor, error reduction techniques, strain gauge arrangement, least squares method

## Abstract

In this study, a novel strain gauge arrangement and error reduction techniques were proposed to minimize crosstalk reading and simultaneously increase sensitivity on a decoupled six-axis force–moment (F/M) sensor. The calibration process that comprises the least squares method and error reduction techniques was implemented to obtain a robust decoupling matrix. A decoupling matrix is very crucial for minimizing error and crosstalk. A novel strain gauge arrangement that comprised double parallel strain gauges in the decoupled six-axis force–moment sensor was implemented to obtain high sensitivity. The experimental results revealed that the maximum calibration error, F/M sensor measurement error, and crosstalk readings were reduced to 3.91%, 1.78%, and 4.78%, respectively.

## 1. Introduction

Six-axis force–moment (F/M) sensors have been widely used in robot-based intelligent automation. Most of these commercial sensors are considerably expensive [1,2,3]. These devices can simultaneously measure three axial forces (*Fx*, *Fy*, and *Fz*) and three axial moments (*Mx*, *My*, and *Mz*). The force and moment components between the last link of the robot and its end effector are transduced to a strain gauge by deflecting integrated sensor-elastic elements and by converting mechanical strain into electrical signals. This sensor design can be attached to different grippers or end effectors of robotic systems.

In general, six-axis F/M sensors are categorized into two types based on the relationship between the output signal and applied force: mechanically coupled sensors and mechanically decoupled sensors. Mechanically coupled sensors can generate more than one signal or bridge circuit when a pure force or moment is applied. These sensors require a complicated decoupling algorithm. Conversely, mechanically decoupled sensors can generate a single signal on a bridge circuit when a pure force or moment is applied. The primary advantage of mechanically decoupled sensors is that they do not require a complicated decoupling algorithm [4,5], and the sensors enable the subtraction of sensor components without structural modification. These sensors have the following disadvantages: they have a complex design geometry and generate a large amount of crosstalk readings compared with mechanically coupled force sensors. Therefore, mechanically decoupled sensors are not widely used in industrial applications.

Many researchers have developed a Maltese-type force sensor, which is a decoupled force sensor [1,3,5,6,7,8,9] with different measurement characteristics [10,11,12]. Different types of strain gauges, arrangements, and decoupling algorithm methods provide different results. The actual results of six-axis F/M sensors are unsatisfactory in various aspects because of the decoupling algorithm methods used and strain gauge arrangements. In previous studies, strain gauge positioning has failed to completely utilize the maximum sensitivity achieved by sensors during the measurement of *Mx* and *My*.

The accuracy of the six-axis F/M sensors is strongly dependent on the decoupling matrix, force sensing element arrangements, and error reduction. Sensor calibration procedures must be accurate and precise in collecting signal output voltage data from a known load when the decoupling algorithm is determined. Force sensing element arrangement can be improved by placing strain gauges at specific positions to obtain high sensitivity. However, two primary error sources exist: the design limitation of the sensor structure and noise signals present in the sensed information [13]. Basic sensor error information can be eliminated to an extent by using a suitable mechanical design and decoupling algorithm, as described in the literature [1,5,13,14]. 

Sensor calibration is tedious, particularly when organizing calibration data, because several readings must be taken from the sensor under precise loading conditions based on gravity. A least squares (LS) method has been generally used to combine these readings into a best fit decoupling matrix [15,16,17], which results in a faster and more accurate calibration procedure [18,19]. 

This study developed an error reduction technique for a decoupled six-axis F/M sensor using a novel strain gauge arrangement. In this study, instead of using a single strain gauge, a double parallel-type strain gauge placement was employed. In this arrangement, all the strain gauges were attached at the locations of an elastic rectangle beam to obtain maximum sensitivity. Furthermore, a decoupling matrix was obtained by applying the LS method and error reduction technique within the calibration process.

The paper is organized as follows. Section 2 briefly presents the structural design and strain gauge arrangement of the six-axis F/M sensor developed in the study. Section 3 describes the sensor calibration procedures and the steps for obtaining the decoupling matrix of the proposed sensor. In Section 4, error reduction techniques are explained, and experimental results of the decoupled six-axis F/M sensor are discussed in Section 5. Finally, Section 6 presents some conclusions.

## 2. Structural Design and Strain Gauge Arrangement

This section presents the structural design and strain gauge arrangement for a decoupled six-axis F/M sensor that was utilized to obtain the desired output of the force and moment components.

### 2.1. Structural Design

The development of the proposed six-axis force sensor was based on the structural design principles proposed in Reference [1]. Figure 1 displays the developed design force sensor. An aluminum alloy (7075-T6) was selected for fabricating the sensor. Furthermore, a modified Maltese cross-element with a thin-plate structure was utilized to improve the sensitivity of force measurement and reduce crosstalk. Figure 1 displays that the mid-section rectangular box was placed in the load application area. Four elastic rectangular beams around the box were connected using an outer ring flange. The connecting part between the elastic rectangular beam and rim (sensor housing) was sufficiently thin to be used as a thin-plate structure.

### 2.2. Strain Gauge Arrangement

In this study, a novel positioning of strain gauges was proposed in which 16 double parallel-type strain gauges were placed on a sensor structure. The strain gauge from HBM (DY43-3/350) with a resistance of 350 Ω was used as a transducer. Figure 2 illustrates the arrangement of 16 double parallel strain gauges. Each set of four strain gauges was arranged into six Wheatstone full-bridge circuits. The placement process involved the attachment of strain gauges at the maximum deformation locations for maximizing sensitivity along the applied force and moment axes. Therefore, the strain gauges were arranged with different force and moment components during measurement. For example, Equation (1) illustrates the application of the bending beam with a force in the *x*-axis of a bridge mounted in tension (SG_28_ and SG_15_) and other mounted in compression (SG_31_ and SG_12_). Moreover, the *x*-axis of a bridge of the bending beam was loaded with a moment mounted in tension (SG_25_ and SG_13_) and the other mounted in compression (SG_30_ and SG_10_). In this full-bridge configuration, Figure 2 presents strain gauge bonding, the output signal of six components of the forces and moments can be measured using strain gauges attached to the beams.
(1)$${\mathrm{SG}}_{\mathrm{Fx}}=\left(\frac{1}{4}\right)\left(\left({\mathrm{SG}}_{28}-{\mathrm{SG}}_{31}\right)+({\mathrm{SG}}_{15}-{\mathrm{SG}}_{12}\right))\;\;{\mathrm{SG}}_{\mathrm{Fy}}=\left(\frac{1}{4}\right)\left(\left({\mathrm{SG}}_{8}-{\mathrm{SG}}_{3}\right)+({\mathrm{SG}}_{19}-{\mathrm{SG}}_{24}\right))\;{\mathrm{SG}}_{\mathrm{Fz}}=\left(\frac{1}{4}\right)\left(\left({\mathrm{SG}}_{1}-{\mathrm{SG}}_{6}\right)+({\mathrm{SG}}_{18}-{\mathrm{SG}}_{21}\right))\;{\mathrm{SG}}_{\mathrm{Mx}}=\left(\frac{1}{4}\right)\left(\left({\mathrm{SG}}_{25}-{\mathrm{SG}}_{30}\right)+({\mathrm{SG}}_{13}-{\mathrm{SG}}_{10}\right))\;\;{\mathrm{SG}}_{\mathrm{My}}=\left(\frac{1}{4}\right)\left(\left({\mathrm{SG}}_{17}-{\mathrm{SG}}_{22}\right)+({\mathrm{SG}}_{5}-{\mathrm{SG}}_{2}\right))\;{\mathrm{SG}}_{\mathrm{Mz}}=\left(\frac{1}{4}\right)\left(\left({\mathrm{SG}}_{32}-{\mathrm{SG}}_{27}\right)+({\mathrm{SG}}_{16}-{\mathrm{SG}}_{11}\right))$$

### 2.3. Finite Element Analysis

In this study, the novel SG arrangement was analyzed using the finite element method. The finite element analysis (FEM) analysis results revealed the sensitivity of the elastic rectangular beam pertaining to the strain gauge placement position. The development of the proposed six-axis force sensor was based on the structural design principles proposed in Reference [1]. The placement of the strain gauges should consider maximum strain, isotropy of the involved material, and prevention of non-linearity [1]. Figure 3a,c,e presents the von Misses strain output for a force of 100 N along the *x*, *y*, and *z*-axes, respectively. Figure 3b,d,f illustrates the von Misses strain output for a given moment of 14 Nm in the *x*, *y*, and *z*-axes, respectively.

Finite element analysis was conducted on an elastic rectangular beam to obtain the exact position for strain gauge placement. Figure 4 shows the non-linearity of strain that occurs near both ends of the elastic rectangular beam. By starting at 3 mm from the rectangular box, the normal strain on the beam was observed to decrease linearly. To achieve high sensitivity and avoid non-linearity, a double parallel-type strain gauge was placed at a position of 4 mm. The current sensors that can be found in the literature often have 24 strain gauges on the cross beam. All strain gauges of force components are 4 mm away and those of moment components are 7 mm away from the wall of the rectangular box. This implies that strain gauge positioning has failed to completely utilize the maximum sensitivity achieved by sensors during the measurement of *Mx* and *My* [1,3,5,6,7,8,9]. 

Based on the proposed strain gauge arrangement and placement, finite element analyses were conducted to present the sensitivity or strain output while applying loads in different axes. Table 1 presents the crosstalk readings obtained for the novel SG arrangement. When a load was applied in the *Fx* direction, the largest crosstalk reading was observed in SG_My_, which was approximately 10%. The arrangement in SG_Mx_ and SG_Mz_ could cancel each other under the *Fx* load. Therefore, the readings in SG_Mx_ and SG_Mz_ were 1% and 0%, respectively. However, the strain gauges of SG_My_ should not be excessively deformed because of the thin-plate structure.

When applying a load in the *Fy* direction, the largest crosstalk reading was observed in SG_Mx_ at approximately 11%. In this arrangement, SG_Fz_ and SG_My_ could cancel each other under the *Fy* load. Therefore, the readings of SG_Fz_ and SG_My_ were 0% and 1%, respectively. However, the strain gauges of SG_Mx_ should not be excessively deformed because of the thin-plate structure.

By contrast, loading in the *Fz* direction yields very low crosstalk readings that only occur in SG_My_ and SG_Mz_. This phenomenon was observed because in this arrangement, SG_Fx_, SG_Fy_, SG_Mx_, SG_My_, and SG_Mz_ could largely cancel each other when the entire beams were deformed. 

When applying a load in the *Mx* direction, the largest crosstalk reading was observed in SGFy, which was approximately 7%. In this arrangement, SG_Fx_ and SG_Mz_ could cancel each other under the *Mx* load. Therefore, readings in SG_Fx_ and SG_Mz_ were 0%. However, the strain gauges of SG_Fy_ were slightly deformed because of the twisting moment.

When a load was applied in the *My* direction, the largest crosstalk reading was observed in SG_Fx_ at approximately −8%. In this arrangement, SG_Fy_ and SG_Fz_ could cancel each other under the *My* load. Therefore, the readings in SG_Fy_ and SG_Fz_ were 2% and 1%, respectively. The strain gauges of SG_Fx_ should be slightly deformed because of the twisting moment.

By contrast, under the *Mz* loading, low crosstalk readings were obtained, except in SG_Fz_, which showed a crosstalk reading of approximately −6%. This phenomenon was observed because in the mentioned arrangement, SG_Fx_, SG_Fy_, SG_Mx_, and SG_My_ cancel each other when all beams are deformed together.

### 2.4. Measurement Principle

Various methods are available for detecting forces and moments by using an electrical circuit measurement technique, and strain gauges are used as force sensing resistors. In a recognized measurement technique for strain gauge elements, electrical resistance is proportional to the amount of strain. When strain gauges are bonded on an elastic element of the sensor, they can detect changes in the resistance after deformation of the force sensing element. A gauge factor (GF), which is a fundamental parameter for measuring the sensitivity to strain, can be expressed as follows:(2)$$\mathrm{GF}=\frac{\Delta R/R}{\Delta L/L}=\frac{\Delta R/R}{\varepsilon }$$
where *R*, *L*, and *ε* represent the original resistance of the strain gauge, original length, and strain representing a fractional change in length, respectively. The GF of metallic strain gauges is generally approximately 2.

Strain gauges configured in the bridge circuit are used to measure small changes in resistance. Full-bridge circuits with four arms used as active strain gauges can further improve the sensitivity of the circuit, and the measurement sensitivity of the circuit can be represented as follows:(3)$$\frac{V_{o}}{V_{E}}=-\mathrm{GF}.\varepsilon$$
where $V_{o}$ and $V_{E}$ are the output voltages of the bridge circuit and voltage excitation source, respectively.

## 3. Sensor Calibration 

In multi-axis force sensors, the performance of sensing sensors is strongly based on the decoupling matrix, force sensing element arrangements, and error reduction techniques. This method is crucial for enhancing the accuracy of the F/M sensor.

### 3.1. Decoupling Matrix

In the calibration operation, standard weights are directly placed on the sensor mounted on the calibration jig, and then, the sensor elements generate a number of voltage values. The standard technique for solving the calibration problem is used to determine the linearity of the force sensor measurement system by applying the LS or pseudoinverse method. The LS and pseudoinverse methods are equivalent [20]. Known loads and the measurement of corresponding sensor output voltages are required to collect the calibration data. The decoupling matrix can be obtained through the calibration operation.

A multi-axis force sensor is a device in which several simple transducers measure the effects of unknown loads (measured force) on the linear elastic behavior of a mechanical structure. Consider a structure loaded (input) at a particular point by using a measured force $\vec{F}=\left(F_{x},F_{y},F_{z},M_{x},M_{y},M_{z}\right)$. When a pure input load is applied in the *x* direction of the F/M sensor $\vec{F_{1}}=\left(f_{11\;}0\;0\;0\;0\;0\right]{)}^{T}$, multiple components are generated in an output signal from the sensor $\vec{V}=\left(v_{1},v_{2},v_{3},\ldots ,v_{n}\right)$ at the prescribed locations. The linear elasticity relationship can be expressed using Equation (4), as follows:(4)$$\vec{V}=\left[C\right].\vec{F}$$
where [C] is a compliance matrix in which entries in each column are strains induced by an applied force in the sensor output voltage for each corresponding force component.

Entry *c_ij_* denotes the strain contribution in the *i^th^* sensor output voltage generated by the application of a known load of the *j^th^* force or the moment component. By considering six output signal measurement locations (i.e., *n* = 6), [C] is a 6 × 6 diagonal matrix, and the solution of the measured force vector is directly obtained through a matrix inversion operation by using Equation (5), as follows:(5)$$\vec{F}={\left[C\right]}^{-1}.\vec{V}$$

The compliance matrix [C] vectors are as follows:(6)$$C=\left[\begin{array}{ccc} c_{11} & c_{12} & \begin{array}{cc} \ldots & c_{16} \end{array}\\ \begin{array}{c} c_{22}\\ \begin{array}{c} \vdots \\ c_{61} \end{array} \end{array} & \begin{array}{c} c_{22}\\ \begin{array}{c} \vdots \\ c_{62} \end{array} \end{array} & \begin{array}{c} \begin{array}{cc} \ldots & c_{26} \end{array}\\ \begin{array}{c} \begin{array}{cc} \ddots & \vdots \end{array}\\ \begin{array}{cc} \ldots & c_{66} \end{array} \end{array} \end{array} \end{array}\right]$$

[C]^−1^ is the inverse compliance matrix of the sensor that directly multiplies the output signal of the sensor to obtain a decoupled F/M measurement. For *n* higher than six, direct inversion cannot be performed; therefore, the force vector is evaluated using a pseudoinverse technique of operating matrices [20]. Therefore, the force vector is obtained using Equation (7), as follows:(7)$$\vec{F}={\left({\left[C\right]}^{T}\left[C\right]\right)}^{-1}{\left[C\right]}^{T}\vec{V\;}=\left[D\right].\vec{V}$$
where the decoupling matrix [D] can be obtained using the pseudoinverse technique of operating matrices. Operating matrices comprise the compliance matrix obtained by applying the LS method in the calibration process. 

### 3.2. Calibration Setup

The calibration procedure of the sensor was performed to determine the decoupling matrix [D]. Most decoupled six-axis wrist F/M sensors have some crosstalk between their F/M components. Therefore, an accurate calibration process is required.

In the calibration setup, F/M sensors are the source of the reference force or reference moment. In general, calibration is performed using a standard weight or a set of standard weights to produce a reference force under gravitational pull. The reference moment can be established using a known distance lever and reference force on the calibration jig. Determining the actual value of the gravitational constant at the location of the calibration device may be necessary for special applications. In this study, four standard weights were applied in the positive and negative directions under unloaded conditions. The calibration test was performed by applying pure forces and moments without any eccentricity (Figure 5). Figure 5a displays the calibration setup for the loading test of pure axial forces *Fx* and *Fy*. The developed sensor was mounted at the center of the calibration jig and then tightened from the top side of the cross lever of the jig. Figure 5b displays the calibration setup for the loading test of the pure axial force *Fz*. Figure 5c,d presents the calibration setups for the loading test of the pure moments *Mx*, *My*, and *Mz*. The signal voltage output of the SG was calculated using Equations (1) and (3).

The calibration procedure is as follows:The zero point of the output signal of six filtered channels is adjusted using the error reduction technique.Pure forces and moments are applied to the mounted six-axis F/M sensor on the calibration jig through hanging weights with a 2.5 kg increment from +10 kg to −10 kg.The output voltage signal response is measured and counted for each data point corresponding to the applied load of the six-channel force sensor, and each axis of the measured force and moment is generated.The load is applied for 100 s, and the output voltages in each component are collected by taking the calculated average value.After completing the applied load, the decoupling matrix [D] is obtained using the pseudoinverse method. The output signals of the sensor and decoupling matrix are used to calculate the measured forces and moments.

## 4. Error Reduction Techniques

Error in measuring tools is inevitable and makes measuring and experimental processes uncertain. Saturation, noise, drift, and hysteresis are common errors on the six-axis F/M sensor. Temperature, mechanical coupling, electrical coupling, or internal failure causes these errors. It is well known that Wheatstone full bridge is used in the strain gauge arrangement to compensate for errors caused by temperature changes such as saturation and drift. In this research, full-bridge arrangement was used to increase sensitivity and reduce errors. Therefore, to manage such errors, the error reduction techniques of an accurate force sensing system should be developed (Figure 6). 

### 4.1. Force Filtering

Noise, on the other hand, could be handled by one of the digital filters. It is well known to use a low-pass filter for six-axis force sensor with the advantage of removing inner, high-frequency structural resonances of the transducer while maintaining its undistorted low-frequency signal region of concern. This eliminates the possibility of aliasing where high-frequency data “folds over” and corrupts this so-called low-frequency signal region due to the insufficient sampling rate, and more efficiently uses the data bandwidth and storage capability of the measuring system [21].

In order to evaluate the best uses of filtering type, experiments are needed to try all of the effects. Four types of filtering techniques that provided by LabView, Inverse Chebyshev, Butterworth, Chebyshev, and Elliptic were compared to give the best result. As shown on Figure 7, Inverse Chebyshev eliminate ripple give smooth amplitude and fast response time.

### 4.2. Moving Average Filter

Drift is another type of error that results from changes in temperature, mechanical coupling, or internal failure. Wheatstone’s complete bridge circuit addressed the handling error triggered by temperature modifications. However, the moving average filtering was performed for other causes to minimize the errors. We named this differently in two different processes. We call it voltage zero in the calibration process, and we named it offset in the measurement process.

#### 4.2.1. Voltage Zero

This process assumes the start of the calibration process. We need to obtain filtered input, free from drift, to get a robust matrix of decoupling. Although there is no loading, while we attach the F/M sensor to the calibration jig, it generates some voltage reading (drift). Voltage zero is intended to zero these readings.

#### 4.2.2. Force Offset

Different from voltage zero, force offset is needed in the measurement process. When we attach the F/M sensor on the working apparatus and with the tool, obviously it will create force reading but with drift that is caused by mechanical coupling among them. These errors would be eliminated by average filtering that we called force offset. 

### 4.3. Error Reduction Techniques Process

#### 4.3.1. Calibration Process

The error reduction techniques that were used within the calibration process are as follows:First step: the signals from the F/M sensors should be filtered using “force filtering” techniques.Second step: after installing F/M into the calibration jig, before loading, “voltage zero” technique should be used to set six readings to zeros.

#### 4.3.2. Measurement Process

The error reduction techniques to be used after the calibration process comprises the measurement process as follows:First step: to get the filtered output signal from F/M sensors using “force filtering” techniques.Second step: after installing F/M into the end effector and installing the tool on it, before the start of robot applications, the “force offset” process should be conducted to set six undesirably existing readings to zero.

## 5. Experimental Results and Discussion

### 5.1. Software Configuration

The six-axis force–moment sensors comprise six Wheatstone bridges and configured data acquisition (DAQ) programs that provide users control over all important experimental parameters. This decoupled sensor was electronically connected using LabVIEW 2014 (National Instruments, Austin, TX, USA). A serial communication port of the PCI-6229 card was interfaced to a computer with LabVIEW 2014/National Instruments (NI) using Windows XP. LabVIEW is an object-oriented software programming tool that helps visualize various application aspects, including hardware settings, measurement data, and debugging. The LabVIEW process is divided into two sections, namely, a front panel and block diagram. The front panel is visible when a program runs and is similar to any graphical user interface. Moreover, it comprises controls for input operations and indicators for output operations and contains icons that represent operators for block diagram programming. The primary channel outputs of the six-axis F/M sensor were captured using LabVIEW. 

The PCI-6229 multifunction data acquisition card (National Instruments) is effective in measuring analog output channels and features 32 single-ended or 16 differential channels. The resolution is 16-bit and with a 250 kS/s sample rate.

### 5.2. Experimental Results

In this study, a six-axis calibration jig was developed to apply pure forces and moments on different axes. The output voltage signal responses of all six full-bridge circuits were directly measured using the National Instrument NI-DAQ after configuring the implemented error reduction techniques in LabVIEW. Several experimental data were obtained in the experimental test of the decoupled six-axis F/M sensor through the data acquisition system. The experimental test was similar to the calibration data collection process. Pure forces and moments were applied to the mounted six-axis F/M sensor on the calibration jig through hanging weights with a 2.5 kg increment from +10 kg to −10 kg. In our experiment, one hundred samples were taken for every single load given. A standard weight was used as the load, and the weight was transformed using a wire and pulley into forces and moments. The moments (*Mx* and *My*) at 0.1358 m and (*Mz*) at 0.1 m were referenced using a known distance lever in the reference force of the calibration jig. The decoupling matrix on the F/M sensor was required to convert the signal voltage to Newton.

The low-pass filter was designed with a cutoff frequency of 50 Hz for the optimal finite impulse response. The data can be obtained over a certain restricted range of parameters for the desired output results. The cutoff frequency transformed the received pilot signal data using a selected filter order. While filtering, the input signal was stabilized within 0.4 s to obtain the filtered output signal (Table 2).

The accuracy of the decoupling matrix was crucial for improving the precision of the six-axis F/M sensor. The proposed error reduction techniques and novel SG arrangement were used to improve sensor accuracy. A decoupling algorithm of the six-axis force sensor was proposed based on the calibration error and sensor F/M measurement error. Furthermore, to improve sensor accuracy, the sensor design allowed loading in the positive and negative directions. As shown in Table 3 and Table 4, the experimental result calibration–interference errors between the best fit values of the LS method and maximum output voltage signal were 3.91% and −3.1% in the positive and negative directions, respectively. Furthermore, Table 5 presents the sensor F/M measurement error of 0.61% in *Fx*, 0.37% in *Fy*, 0.58% in *Fz*, 1.78% in *Mx*, 1.74% in *My*, and 1.45% in *Mz*. However, the maximum crosstalk readings on the applied load components of *Fx*, *Fy*, *Fz*, *Mx*, *My*, and *Mz* were 1.03%, 1.44%, 3.36%, 4.78%, 3.48%, and 1.28%, respectively. Therefore, the maximum crosstalk reading was 4.78% on the *Mz* axis. 

The existing decoupled type F/M sensor error analysis on the decoupling algorithm based on LSM in Reference [22] indicates a F/M Measurement error and a crosstalk error of 6.85% and 4.85% respectively. The error reduction techniques of the decoupled six-axis F/M sensor significantly improved the sensor output signals and minimized erroneous readings of the measured force and moment. Finally, the experimental test results of this error reduction implementation process validated the excellent minimization of the calibration error and sensor F/M measurement error based on the decoupling matrix.

## 6. Conclusions

A decoupled six-axis F/M sensor employs several techniques to optimize the accuracy of the measured force and moment. The LabVIEW environment was used to perform signal processing and to develop a compensation algorithm. A novel SG arrangement and error reduction techniques were developed on the physical system. In order to reduce the effect of the common errors signal to sensor accuracy, all the software and hardware were properly configured. The signal process of the analog voltage signal was converted to digital numbers, the digital signal filtered by the second order Inverse Chebyshev low-pass filter, while retaining a sharp step response. Additionally, the moving average filter was optimal for the crucial tasks of voltage zero and force offset to reduce random noise. The validity of the proposed methods were evaluated experimentally using a six-axis F/M sensor.

The accuracy of the developed six-axis F/M sensor was excellent. This behavior was partially attributed to the precise alignment of the SG and elastic element. The sensor employed a calibration system and decoupling algorithm with compensation, which in turn employed the LS method. These decoupling algorithms and error reduction techniques were developed in LabVIEW. The maximum calibration error, sensor F/M measurement error, and crosstalk readings were 3.91%, 1.78%, and 4.78%, respectively. The results were obtained with relatively unsatisfactory readings in *Mx* and *My* only. Moreover, the results reveal excellent accuracy characteristics, such as high F/M measurement sensitivity, suitable (calibration) linearity, and low-noise signal sensor information. The error reduction techniques can help users to easily manage the complicated calibration system and improve sensor accuracy. This study can be beneficial for further research into the practical usage and industrial applications of decoupled six-axis F/M sensors.

## Figures and Tables

**Figure 1 sensors-19-03012-f001:**
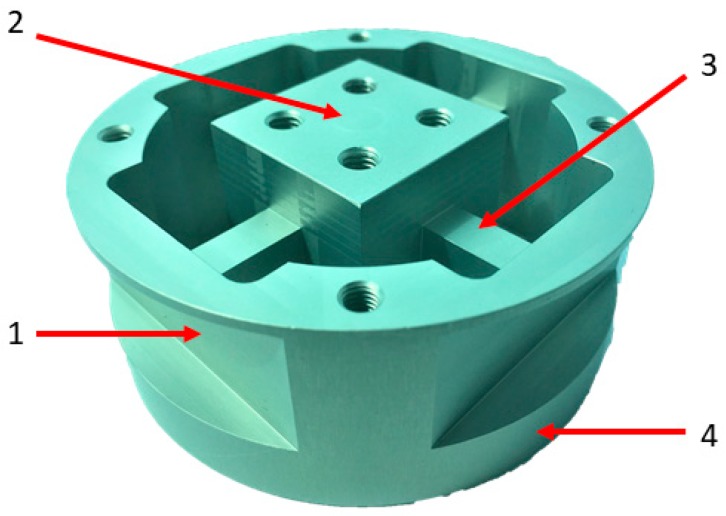
Structural design of the proposed force sensor: (**1**) thin plate, (**2**) rectangular box, (**3**) elastic rectangular beams, and (**4**) rim (sensor housing).

**Figure 2 sensors-19-03012-f002:**
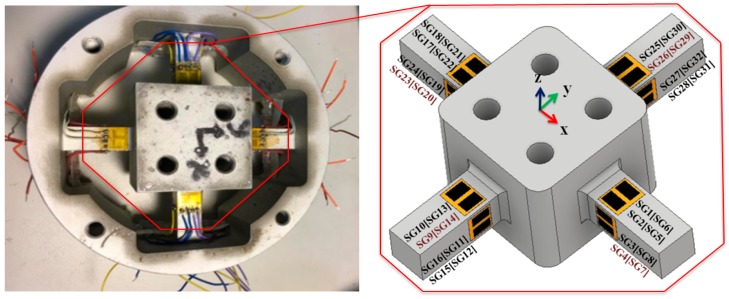
Strain gauge arrangement: (SG) active strain gauge and (SG) idle strain gauge.

**Figure 3 sensors-19-03012-f003:**
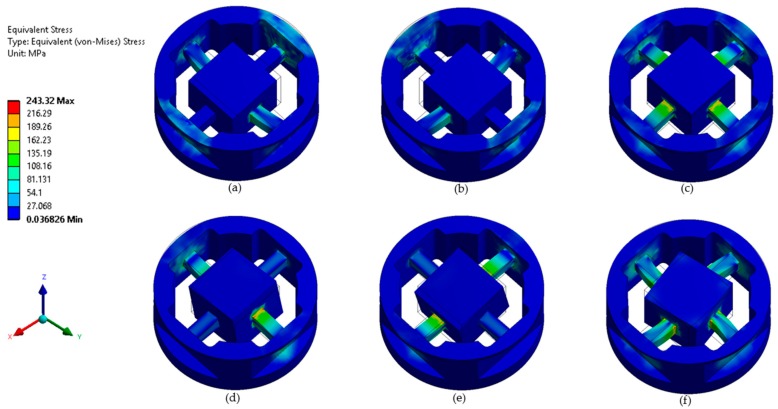
Finite element analysis: (**a**) pure load *Fx*, (**b**) pure load *Fy*, (**c**) pure load *Fz*, (**d**) pure load *Mx*, (**e**) pure load *My*, and (**f**) pure load *Mz*.

**Figure 4 sensors-19-03012-f004:**
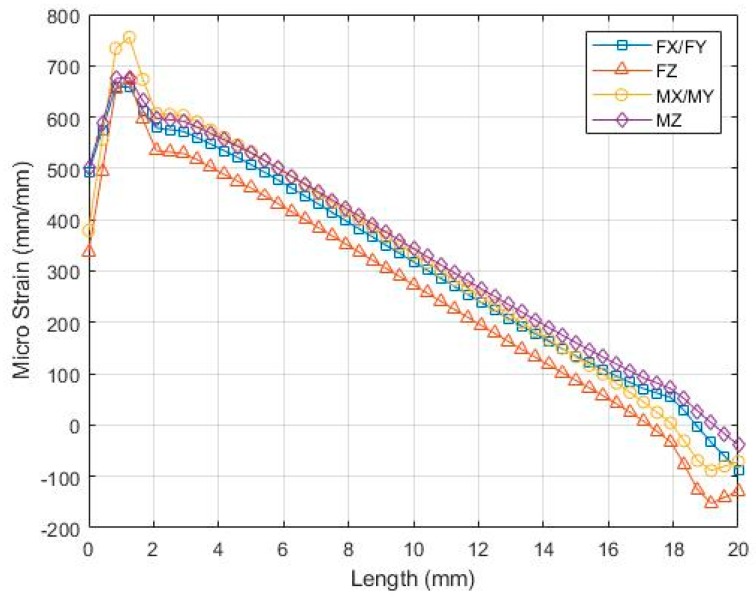
Normal strain on the middle rectangular beam.

**Figure 5 sensors-19-03012-f005:**
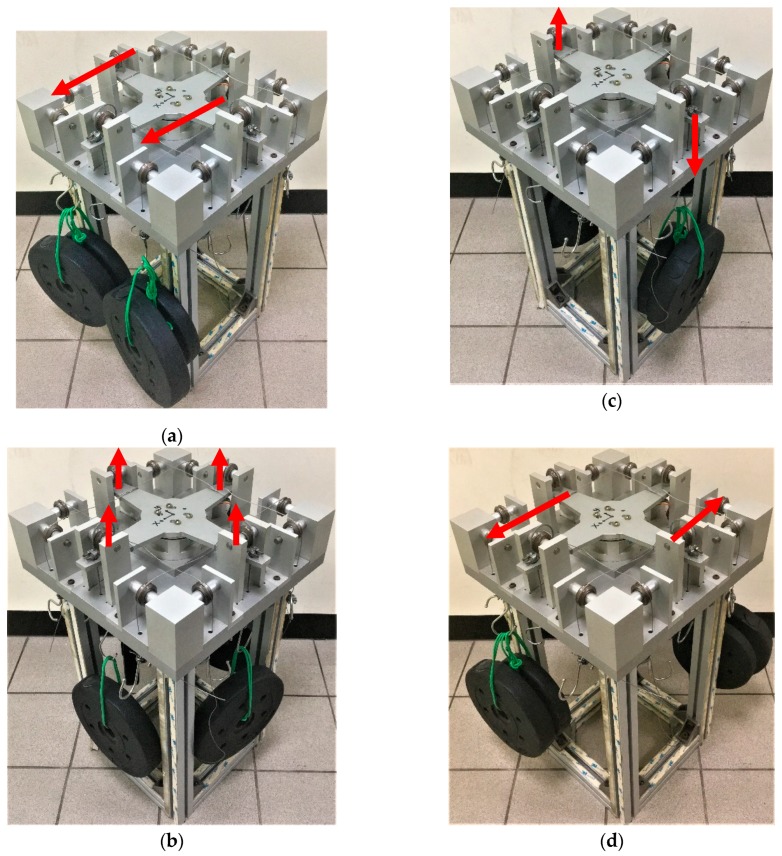
Schematic expression of the applied load conditions on the jig: (**a**) *Fx* or *Fy*, (**b**) *Fz*, (**c**) *Mx* or *My*, and (**d**) *Mz*.

**Figure 6 sensors-19-03012-f006:**
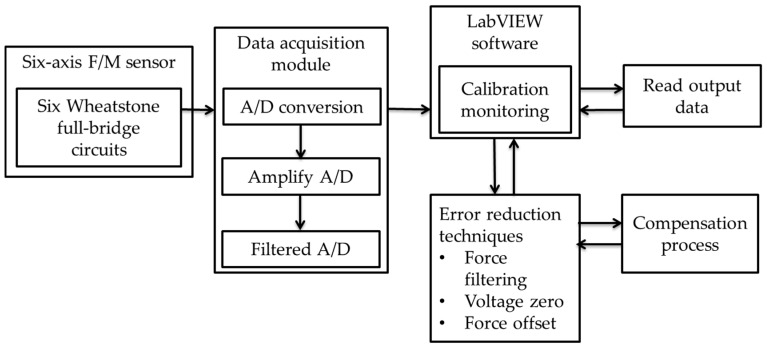
Module and process of a six-axis force–moment (F/M) sensor.

**Figure 7 sensors-19-03012-f007:**
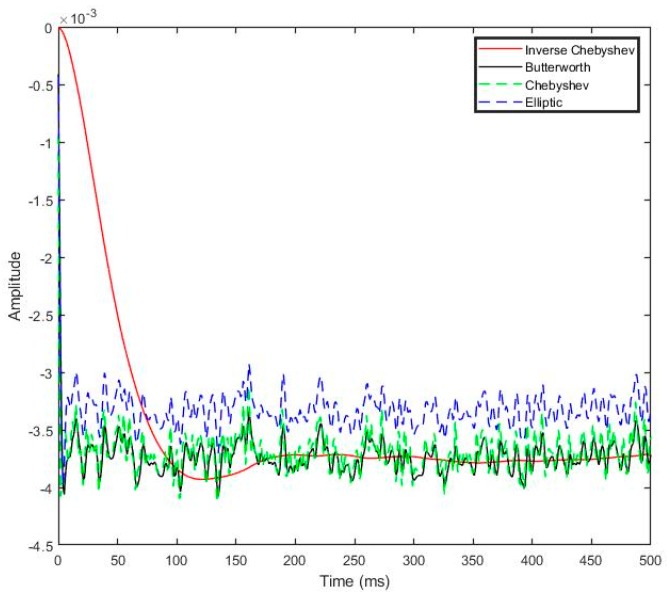
The response of filtering output systems on the four different filter types.

**Table 1 sensors-19-03012-t001:** Crosstalk readings of the finite element analysis results.

Crosstalk Reading (%)		Axis of Reading					
		*Fx*	*Fy*	*Fz*	*Mx*	*My*	*Mz*
**Applied load**	*Fx*	-	0%	0%	1%	−10%	0%
	*Fy*	−1%	-	0%	11%	1%	0%
	*Fz*	0%	0%	-	−1%	0%	-6%
	*Mx*	0%	7%	0%	-	0%	0%
	*My*	−8%	2%	1%	0%	-	0%
	*Mz*	0%	0%	−1%	−1%	0%	-

**Table 2 sensors-19-03012-t002:** Inverse Chebyshev II low-pass filter signal output.

Input Signal Components	Impulse Signals Response Range (V)		Filtered Output Signal (V)
*Fx*	upper	−0.0037	−0.00445
	lower	−0.0052	
*Fy*	upper	0.00125	0.00028
	lower	−0.0007	
*Fz*	upper	−0.0086	−0.00935
	lower	−0.0101	
*Mx*	upper	0.00075	−0.00058
	lower	−0.0019	
*My*	upper	0	−0.00095
	lower	−0.0019	
*Mz*	upper	0.0028	0.00175
	lower	0.0007	

**Table 3 sensors-19-03012-t003:** Experimental results of the calibration errors in the sensor positive measurement direction under the maximum load.

Sensor Components	LS Best Fit Value (V/V)	Experimental Value (V/V)	Error (%)
*Fx* = 98.1 N	0.077	0.075	2.40
*Fy* = 98.1 N	0.089	0.090	0.80
*Fz* = 98.1 N	0.055	0.056	1.00
*Mx* = 13.322 Nm	0.336	0.323	3.91
*My* = 13.322 Nm	0.433	0.432	0.28
*Mz* = 9.81 Nm	0.153	0.154	0.45

**Table 4 sensors-19-03012-t004:** Experimental results of the calibration errors in the sensor negative measurement direction under the maximum load.

Sensor Components	LS Best Fit Value (V/V)	Experimental Value (V/V)	Error (%)
*Fx* = −98.1 N	−0.077	−0.079	−3.10
*Fy* = −98.1 N	−0.094	−0.093	−0.55
*Fz* = −98.1 N	−0.060	−0.060	−0.19
*Mx* = −13.322 Nm	−0.340	−0.343	−0.84
*My* = −13.322 Nm	−0.421	−0.413	−1.95
*Mz* = −9.81 Nm	−0.151	−0.147	−2.71

**Table 5 sensors-19-03012-t005:** F/M measurement error and crosstalk reading in the experimental results of the sensor.

F/M Measurement Error and Crosstalk Reading (%)		Axis of Reading					
		*Fx*	*Fy*	*Fz*	*Mx*	*My*	*Mz*
**Applied load**	*Fx*	0.61	0.53	1.01	0.57	0.33	1.03
	*Fy*	0.77	0.37	0.81	0.45	0.32	1.44
	*Fz*	1.77	1.78	0.58	3.36	0.40	3.31
	*Mx*	1.71	0.98	1.86	1.78	0.31	4.78
	*My*	1.17	1.47	1.39	0.79	1.74	3.48
	*Mz*	1.23	1.24	1.28	0.68	0.12	1.45

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
