# Peer review of "Decoupled Six-Axis Force–Moment Sensor with a Novel Strain Gauge Arrangement and Error Reduction Techniques"

_sensors, 2019, doi:10.3390/s19133012_

Round 1
Reviewer 1 Report
The development of a six-axis F/T sensor with a new strain-gauge arrangement and a calibration method. They claim an original calibration method in real-time.
In general, the style of the manuscript must be improved before content evaluation.
In particular, the abstract and the introduction sections are confusing, ambiguous, and hard to understand or misleading.
The title is also confusing. I don't see "Real-time calibration" anywhere. The calibration process is done off-line with a laboratory setup.
As far as I understood, the calibration method based on least squares is not new. If so, please clarify this and summarize the advantages.
The Strain gauge arrangement is based on the results of a FEM simulation based on a predefined geometry. It is not clear how this design has been obtained or the difference with current sensors.
Quality of the equations style can be improved.
Figure 3 is taking a full page with screen captures. Legends and axes are redundant here. Backgrounds should be removed.
Figure 4 is not a flow diagram, even if it contains a "start" block. That's a hierarchical system diagram that shows relationships between modules and processes. Block labels are confusing, and the description in the text is not clear.
Figure 5 Is a "Flow chat" so "Start" and "end" blocks would help. Strangely, the decision block output labels have been placed as round blocks. Anyway, this is an almost-sequential process that could be described as a list of steps.
The description of the used language and programming environment is unnecessary and confusing. Figure 6 has cropped screen-captures. The user interface looks like an interface for the developer with debugging purposes only. There is no further use of this figure in the text.
Figure 7 shows a cad design of the calibration setup with poor resolution and wrong aspect ratios. A set of pictures of the real setup is needed instead.
Figure 8 is unacceptable in this journal. Save the data and use proper scientific visualization and exporting software (i.e. Matlab).
In the experiments section, "numerous" doesn't describe precisely the process.
The term "performance" in the manuscript should be replaced by other more specific such as "accuray", "resolution", ...
The term real-time is used in the whole manuscript without a single reference to computing or response time requirements.
Please, to make your work better evaluated, rewrite this manuscript using more concise writing, use proper terms, and focus on your real contribution.
Author Response
Reply to Reviewer 1
Firstly, we would like to thank the reviewer for his in-depth and careful reading of our manuscript entitled “Real-time calibration system for improved performance of a decoupled six-axis wrist force/moment sensor”. We have revised the manuscript and followed the reviewer’s advice as much as possible. We have tried to provide all suitable answers for the questions according to our knowledge.
1. The development of a six-axis F/T sensor with a new strain-gauge arrangement and a calibration method. They claim an original calibration method in real-time.
Answer: Appreciated for pointing that out correctly.
2. In general, the style of the manuscript must be improved before content evaluation.
Answer: We have modified the style of the manuscript as advised.
3. In particular, the abstract and the introduction sections are confusing, ambiguous, and hard to understand or misleading.
Answer: Following reviewer’s comments, we have rewritten the abstract and the introduction sections. Additionally, we are including how we organize the paper explained in lines 70-74.
“The paper is organized as follows. Section 2 briefly presents the structural design and strain gauge arrangement of the six-axis F/M sensor developed in the study. Section 3 describes the sensor calibration procedures and the steps for obtaining the decoupling matrix of the proposed sensor. In Section 4, error reduction techniques are explained, and experimental results of the decoupled six-axis F/M sensor are discussed in Section 5. Finally, Section 6 presents some conclusions.”
4. The title is also confusing. I don't see "Real-time calibration" anywhere. The calibration process is done off-line with a laboratory setup.
Answer: We apologize for the clarification that has been misused. What we meant by the term real-time calibration system, when determining a decoupling matrix during the calibration process we implemented error reduction techniques, upon loading a predetermined weight along an axis we would simultaneously obtain the decoupling matrix. This decoupling matrix would keep changing/ updating until the entire calibration process was done. To avoid misuse of term of real-time calibration we are not using it in the revised paper.
Based on this explanation, we changed the title into “Decoupled six-axis force–moment sensor with a novel strain gauge arrangement and error reduction techniques”. The calibration process that we meant was explained in lines 228-240 and comprised of the error reduction technique defined in lines 248-258 in the revised manuscript to clarify the mentioned comments.
5. As far as I understood, the calibration method based on least squares is not new. If so, please clarify this and summarize the advantages.
Answer: We agree with the reviewer’s comment, a least squares solution is a popular method for applied to decoupling algorithm and we briefly referred to it in [15-17] and [18,19]. Since our research targets a least squares method on calibration process, we combined it with Force Zero technique to obtain the best-fit decoupling matrix. The error in the six-axis force/moment sensor output signals mostly occurs because of the inaccuracy of the sensor structure, which generates a coupling effect in channels and noise signals from several sources. Therefore, to minimize these errors, a decoupling matrix that decouples output signals is required. The calibration process, comprising the least square method and error reduction techniques, of an intelligent force sensing system is a key towards sensor development. The advantage of employing the least squares is it results in a faster and more accurate calibration procedure as stated in line 63.
“Which results in a faster and more accurate calibration procedure.”
6. The strain gauge arrangement is based on the results of a FEM simulation based on a predefined geometry. It is not clear how this design has been obtained or the difference with current sensors.
Answer: To clarify the reviewer’s concerns, we have included this part in the revised manuscript, lines 118-126.
“Finite element analysis was conducted on an elastic rectangular beam to obtain the exact position for strain gauge placement. Figure 4 shows the nonlinearity of strain that occurs near both ends of the elastic rectangular beam. By starting at 3 mm from the rectangular box, the normal strain on the beam is observed to decrease linearly. To achieve high sensitivity and avoid nonlinearity, a double parallel-type strain gauge was placed at a position of 4 mm. The current sensors that can be found in the literature often have 24 strain gauges on the cross beam. All strain gauges of force components are 4 mm away and those of moment components are 7 mm away from the wall of the rectangular box. This implies that strain gauge positioning has failed to completely utilize the maximum sensitivity achieved by sensors during the measurement of Mx and My [1,3,5-9].”
7. Quality of the equations style can be improved.
Answer: We have made modification on equations to improve their style.
8. Figure 3 is taking a full page with screen captures. Legends and axes are redundant here. Backgrounds should be removed.
Answer: Following the reviewer’s comment, we have updated it accordingly.
9. Figure 4 is not a flow diagram, even if it contains a "start" block. That's a hierarchical system diagram that shows relationships between modules and processes. Block labels are confusing, and the description in the text is not clear.
Answer: Following the reviewer’s comment, we have rewritten the block labels accordingly.
10. Figure 5 Is a "Flow chart" so "Start" and "end" blocks would help. Strangely, the decision block output labels have been placed as round blocks. Anyway, this is an almost-sequential process that could be described as a list of steps.
Answer: Based on the reviewer’s comments, we have changed to a list of steps and this part included in the revised manuscript, lines 248-258.
“The six-axis F/M sensor error reduction process with LabVIEW steps are as follows:
Connect the force to the hardware and software setup to detect the sensor output signal.
Filter the output signals.
Set the “force zero” (reset) filtered signal during the unloading condition.
Set the force reference frame (set applied force positions) process to collect calibration data of different sample loads on sampling rate time.
Meanwhile, obtain the decoupling matrix in the process of force reference frame.
After obtaining the decoupling matrix, set the F/M unit by using the force scaling process.
After completing the above sequential steps, check for any drifting force in the calibrated F/M sensor reading and then perform a force offset. In the absence of a drift, proceed with the F/M measurement.”
11. The description of the used language and programming environment is unnecessary and confusing. Figure 6 has cropped screen-captures. The user interface looks like an interface for the developer with debugging purposes only. There is no further use of this figure in the text.
Answer: We thank the reviewer for the valuable suggestion. Based on the reviewer’s comment, we have removed Figure 6.
12. Figure 7 shows a cad design of the calibration setup with poor resolution and wrong aspect ratios. A set of pictures of the real setup is needed instead.
Answer: We thank the reviewer for the valuable suggestion. We have updated it accordingly.
13. Figure 8 is unacceptable in this journal. Save the data and use proper scientific visualization and exporting software (i.e. Matlab).
Answer: Comments are appreciated. Considering also the other reviewer’s comment on this issue, we have removed Figure 8.
14. In the experiments section, "numerous" doesn't describe precisely the process.
Answer: We thank the reviewer for the valuable suggestion. We have described how experiment data was collected in line 307-311.
“Several experimental data were obtained in the experimental test of the decoupled six-axis F/M sensor through the data acquisition system. The experimental test was similar to the calibration data collection process. Pure forces and moments were applied to the mounted six-axis F/M sensor on the calibration jig through hanging weights with a 2.5-kg increment from +10 kg to −10 kg. Hundred samples were taken for every single load given.”
15. The term "performance" in the manuscript should be replaced by other more specific such as "accuracy", "resolution", ...
Answer: We have considered the suggestions. We have replaced the term, performance by accuracy, in the whole manuscript.
16. The term real-time is used in the whole manuscript without a single reference to computing or response time requirements.
Answer: As we mention in the answer in number 4, we avoid the term real-time and explain what we mean. Therefore, response time is not mentioned in the manuscript.
Please, to make your work better evaluated, rewrite this manuscript using more concise writing, use proper terms, and focus on your real contribution.
Answer: We have rewritten this manuscript concisely as suggested using proper terms (e.g. replaced performance with accuracy, numerous …) and have focused on our main contribution which is strain gauge arrangement and error reduction techniques. We also reorganized the whole manuscript.
Reviewer 2 Report
This paper proposed a decoupled six-axis 73 force/moment sensor based on accurate real-time calibration. The arrangement of strain gauges is a new feature. Double parallel-type strain gauge placement was employed to obtain maximum sensitivity. The contents of the paper is convincing.
The reviewer has four inquiries.
<Inquiry 1>
Figure 1, “(1) thin plate” is hard to grasp. The arrow needs to be changed.
<Inquiry 2>
In pages 4-5, LS method is quite popular. The reviewer feels that Eqs. (7)-(13) can be removed.
<Inquiry 3>
Figure 6 can be removed because it does not provide any scientific information.
<Inquiry 4>
Figure 8 can be removed because it does not provide any insightful information. Readers can understand the characteristics of the filter if they see the filter name. The names of filters are important for readers.
Author Response
Reply to Reviewer 2
Firstly, we would like to thank the reviewer for his in-depth and careful reading of our manuscript. We have revised the manuscript and followed the reviewer’s advice as much as possible. We have tried our best to provide all suitable answers for the questions according to our knowledge.
This paper proposed a decoupled six-axis force/moment sensor based on accurate real-time calibration. The arrangement of strain gauges is a new feature. Double parallel-type strain gauge placement was employed to obtain maximum sensitivity. The contents of the paper is convincing.
1. Figure 1, “(1) thin plate” is hard to grasp. The arrow needs to be changed.
Answer: Following the reviewer’s comment, we have updated it accordingly.
2. In pages 4-5, LS method is quite popular. The reviewer feels that Eqs. (7)-(13) can be removed.
Answer: We thank the reviewer for the valuable suggestion. We have removed Eqs. (7)-(13).
3. Figure 6 can be removed because it does not provide any scientific information.
Answer: We thank the reviewer for the valuable suggestion. We have removed Figure 6.
4. Figure 8 can be removed because it does not provide any insightful information. Readers can understand the characteristics of the filter if they see the filter name. The names of filters are important for readers.
Answer: Based on the reviewer’s comments, we have removed Figure 8.
Round 2
Reviewer 1 Report
I wish to thank the authors their effort in improving the language and style in the abstract and introduction sections, which makes it easier to understand.
I thanks also for the rest of the improvements made.
This paper's contributions focus on two points: A new mechanical and strain-gauge arrangement design, and an "error reduction technique." The design features of the sensor have been clearly described. However, the developed error reduction technique is not clear yet.
In section 4: The term "intelligent" refers to knowledge-based methods, fuzzy, neural-nets, etc. I don't see any of these techniques here. Please clarify the use of this term.
It is not clear the difference between the calibration and measurement processes. The first one is performed once, and the second one is performed once per reading. It looks like the "sensor error reduction process" (line 248) is the calibration process. I don't understand that.
I also don't understand lines 249, 252, and 255.
Section 4.3. I don't understand the section. Rewrite lines 280-281. What is a standard weight? Is this section is called force reference frame because of the physical frame?
The same applies to section 4.4.
Section 4.5 is also unclear. What is the difference between zero and offset? Screw tightening?
Remember that Section 4 is half of the contribution and must be clearly defined.
Please, rewrite section 5.1. Is Software "Configuration" what you mean?. Can a process be divided into sections? Have you described the DaQ system adequately (electronics, sample rate, resolution, etc )?
Section 5.2 "Result" or "Results"?
Line 320 "Desired output signal"?
Tables 3 and 4: Calibration errors are measured comparing the expected voltages (according to the LS method) for the applied forces, and the real ones. However, if the decoupling matrix has been obtained with the same method ¿What is the source of those errors?
How the accuracy of this method compares with existing F/M sensors? Is it similar or way better than existing methods? I don't see any comparison.
Finally, I don't see what is new in the called "error reduction technique." Please, this is something that must be made clear.
By the way, the equations still need to be centered.
Author Response
Firstly, we would like to thank the reviewer for his in-depth and careful reading of our manuscript entitled “Real-time calibration system for improved performance of a decoupled six-axis wrist force/moment sensor”. We have revised the manuscript and followed the reviewer’s advice as much as possible. We have tried to provide all suitable answers for the questions according to our knowledge. We have made the changes in the text by "Track Changes" function in Microsoft Word, which will appear in red color in the manuscript as deletion and blue color as insertion
1. This paper's contributions focus on two points: A new mechanical and strain-gauge arrangement design, and an "error reduction technique." The design features of the sensor have been clearly described. However, the developed error reduction technique is not clear yet.
Answer: Following reviewer’s comments, we have rewritten the error reduction techniques in section 4.
2. In section 4: The term "intelligent" refers to knowledge-based methods, fuzzy, neural-nets, etc. I don't see any of these techniques here. Please clarify the use of this term.
Answer: We apologize for the clarification that has been misused. What we mean to describe the statement in line 258-259,
“Therefore, to manage such errors, the error reduction techniques of an accurate force sensing system should be developed.”
3. It is not clear the difference between the calibration and measurement processes. The first one is performed once, and the second one is performed once per reading. It looks like the "sensor error reduction process" (line 248) is the calibration process. I don't understand that.
Answer: Following reviewer’s comments, we have rewritten the error reduction techniques in section 4. Additionally, we have included calibration process on section 4.6 line 322-330 and measurement process on section 4.7 line 331-340.
4. I also don't understand lines 249, 252, and 255.
Answer: As we mention in the answer in number 3, we have rewritten section 4 to make distinction between calibration process and measurement process.
5. Section 4.3. I don't understand the section. Rewrite lines 280-281. What is a standard weight? Is this section is called force reference frame because of the physical frame?
Answer: To clarify the reviewer’s concerns, the term a “standard weight” we mean “standard weight plates”. We have rewritten section 4.3 to clarify it. The reason of this section called force reference frame because we were using rectangular coordinate system for the sensor.
6. The same applies to section 4.4.
Answer: To clarify the reviewer’s concerns, we have rewritten section 4.4.
7. Section 4.5 is also unclear. What is the difference between zero and offset? Screw tightening?
Answer: To clarify the reviewer’s concerns, we have rewritten section 4.5. The concept of zero and offset are similar, but force zero a technique to reduce errors during the calibration process, exploiting the reading that the sensor output projection through the decoupled matrix. On other hand, force offset option to reduce errors measurement process to obtain an accurate reading during preloading and loading conditions. What we meant screw tightening is when the user didn’t use torque wrench for remount/ assemble the sensor into end effector so stress of screw will be uneven and make some additional stress to the sensor.
8. Remember that Section 4 is half of the contribution and must be clearly defined.
Answer: Comments are appreciated. We have rewritten section 4 clearly in the revised manuscript.
9. Please, rewrite section 5.1. Is Software "Configuration" what you mean?. Can a process be divided into sections? Have you described the DaQ system adequately (electronics, sample rate, resolution, etc )?
Answer: Following the reviewer’s comment, we have rewrite section 5.1 and this part included in the revised manuscript, lines 344-346 and 355-357.
“The six-axis force-moment sensors comprises six Wheatstone bridges and configured data acquisition (DAQ) programs that provide users the control over all important experimental parameters.
The PCI-6229 multifunction data acquisition card (National Instruments) is effective in measuring analog output channels and features 32 single-ended or 16 differential channels. The resolution is 16-bit and with a 250kS/s sample rate.”
10. Section 5.2 "Result" or "Results"?
Answer: We thank the reviewer for the valuable suggestion. We changed “Results” line 360.
11. Line 320 "Desired output signal"?
Answer: To clarify the reviewer’s concerns, what we meant the term desired output signal is filtered output signals, line 377.
12. Tables 3 and 4: Calibration errors are measured comparing the expected voltages (according to the LS method) for the applied forces, and the real ones. However, if the decoupling matrix has been obtained with the same method? What is the source of those errors?
Answer: As we mention in the answer in number 1, we have rewritten section 4 this manuscript to clarify decoupling matrix obtained and source errors.
13. How the accuracy of this method compares with existing F/M sensors? Is it similar or way better than existing methods? I don't see any comparison.
Answer: To clarify the reviewer’s concerns. We have included this part in the revised manuscript, lines 399-401.
“The existing decoupled type F/M sensor error analysis on the decoupling algorithm based on LSM in [21] indicates a F/M Measurement error and a crosstalk error of 6.85% and 4.85% respectively. The error reduction techniques of the decoupled six-axis F/M sensor significantly improved the sensor output signals and minimized erroneous readings of the measured force and moment.”
14. Finally, I don't see what is new in the called "error reduction technique." Please, this is something that must be made clear.
Answer: As we mention in the answer in number 1, We have rewritten section 4 this manuscript to clarify error reduction techniques.
15. By the way, the equations still need to be centered.
Answer: We thank the reviewer for the valuable suggestion. We have updated it accordingly.
Round 3
Reviewer 1 Report
I see some improvements and the manuscript is now more readable.
However, I still have my concerns, mostly about the "Error reduction techniques" section:
- The only error reduction technique I see is "force filtering" but I can't see what's new or special here. Just a mention of the type of filter, without any justification or details (they can be found in the experiments). The last paragraph of this subsection is hard to understand.
- Subsection 4.2 (Force Zero) is just another basic sensor processing operation. What's new?
- Subsection 4.3. I don't understand.
- Subsection 4.4. Obvious.
- Subsection 4.5 (Force Offset) mentions the weight of the end-effector, but nothing about a method. It is obvious that this is not compensated in the decoupling matrix.
- Subsections 4.6 and 4.7 are sequences of the calibration and measurement processes.
My advice is to rethink the structure of the document (or at least section 4).
There are many other language issues.
In its present form, this manuscript is hard to read, and the reader learns nothing.
If this work has relevant content, you must clearly identify tour contribution and the relevance of it. It is necessary, so the reader could find this publication useful and citable.
Author Response
Firstly, we would like to thank the reviewer for his enthusiastic efforts on making this manuscript better and providing in-depth and valuable comments and suggestions. We have carefully revised the manuscript and followed the reviewer’s advice as much as possible. We have tried to provide suitable answers for the questions according to our knowledge. We have made the changes in the manuscript using "Track Changes" function in Microsoft Word.
1. The only error reduction technique I see is "force filtering" but I can't see what's new or special here. Just a mention of the type of filter, without any justification or details (they can be found in the experiments). The last paragraph of this subsection is hard to understand.
Answer: Following reviewer’s comments, we have rewritten and included detailed explanation. The reasons why we considered this type of filter is added in both line 308-317 and 380-384 (line counts in the tracked file). The experimental results of comparison on different filter types are shown in Figure 7.
2. Subsection 4.2 (Force Zero) is just another basic sensor processing operation. What's new?
Answer: To clarify the reviewer’s concern, the term “force zero technique” which we used meant “voltage zero.” This feature reduces error occurring during the phase of calibration by exploiting the reading of the projection of the sensor output. The accuracy of the decoupling matrix will be improved by the implementation of “voltage zero.” We have explained this point in the revised manuscript in line 327-330 (line count in the tracked file).
“This process assumes the start of the calibration process. We need to get filtered input, free from drift, to get a robust matrix of decoupling. Although there is no loading, while we attach the F/M sensor to the calibration jig, it generates some voltage reading (drift). Voltage zero is intended to zero these readings.”
3. Subsection 4.3. I don't understand.
Answer: The force reference frame technique mainly describes how the loading in each of the 6 degree of freedom can be in turn independently applied to the sensor on the jig. Since the calibration process is obviously known in Multi-axis F/M sensors, therefore we have removed the corresponding content in the revised manuscript.
4. Subsection 4.4. Obvious.
Answer: We accept this suggestion and have removed this known calibration process in the revised manuscript.
5. Subsection 4.5 (Force Offset) mentions the weight of the end-effector, but nothing about a method. It is obvious that this is not compensated in the decoupling matrix.
Answer: We agree with the reviewer. The error reduction techniques mainly comprise processes of calibration and measurement. The error of this type cannot be compensated in the decoupling matrix but it can be compensated each time before the sensor is used in different tasks. We have illustrated the content in the revised manuscript in line 332-335 (line count in the tracked file).
“Different from voltage zero, force offset is needed in the measurement process. When we attach the F/M sensor on the working apparatus and with the tool, obviously it will create force reading but with drift that caused by mechanical coupling among them. These errors would be eliminated by average filtering that we called Force Offset.”
6. Subsections 4.6 and 4.7 are sequences of the calibration and measurement processes.
Answer: We thank the reviewer for the valuable suggestion. We have updated the sequences of section 4.6 and 4.7 in the revised manuscript.
7. My advice is to rethink the structure of the document (or at least section 4).
Answer: We thank the reviewer for the valuable suggestion. We have restructured the contents of section 4 and rewrote it in the revised manuscript.
8. There are many other language issues.
Answer: We have modified and proof-read the manuscript carefully with the aid of a native English speaker according to referee’s advice.